# Influence of Metal Salts Addition on Physical and Electrochemical Properties of Ethyl and Propylammonium Nitrate

**DOI:** 10.3390/ijms232416040

**Published:** 2022-12-16

**Authors:** David Ausín, José L. Trenzado, Mireille Turmine, Luis M. Varela, Oscar Cabeza, Elisa González Romero, Luisa Segade

**Affiliations:** 1Departamento de Física y Ciencias de la Tierra, Facultad de Ciencias, Universidade da Coruña, Campus da Zapateira, 15071 A Coruña, Spain; 2Departamento de Física, Universidad de Las Palmas de Gran Canaria, 35017 Las Palmas Gran Canaria, Spain; 3Laboratoire de Réactivité de Surface (LRS), Sorbonne Université, CNRS, 4 Place Jussieu, 75005 Paris, France; 4Grupo de Nanomateriales, Fotónica y Materia Blanda, Departamento de Física de Partículas y Departamento de Física Aplicada, Universidade de Santiago de Compostela, Campus Vida s/n, 15782 Santiago de Compostela, Spain; 5Grupo de Electroanálisis y Biosensores, Departamento de Química Analítica y Alimentaria, Universidade de Vigo, As Lagoas, Marcosende, 36310 Vigo, Spain

**Keywords:** ethylammonium nitrate, propylammonium nitrate, salts, density, viscosity, electrical conductivity, refractive index, surface tension, electrochemical potential windows

## Abstract

In this work, we deepen in the characterization of two protic ionic liquids (PILs), ethylammonium nitrate (EAN) and propylammonium nitrate (PAN). With this aim, we determined the influence of inorganic nitrate salts addition on their physical properties and their electrochemical potential window (EPW). Thus, experimental measurements of electrical conductivity, density, viscosity, refractive index and surface tension of mixtures of {EAN or PAN + LiNO_3_, Ca(NO_3_)_2_, Mg(NO_3_)_2_ or Al(NO_3_)_3_} at a temperature range between 5 and 95 °C are presented first, except for the last two properties which were measured at 25 °C. In the second part, the corresponding EPWs were determined at 25 °C by linear sweep voltammetry using three different electrochemical cells. Effect of the salt addition was associated mainly with the metal cation characteristics, so, generally, LiNO_3_ showed the lower influence, followed by Ca(NO_3_)_2_, Mg(NO_3_)_2_ or Al(NO_3_)_3_. The results obtained for the EAN + LiNO_3_ mixtures, along with those from a previous work, allowed us to develop novel predictive equations for most of the presented physical properties as functions of the lithium salt concentration, the temperature and the water content. Electrochemical results showed that a general order of EPW can be established for both PILs, although exceptions related to measurement conditions and the properties of the mixtures were found.

## 1. Introduction

In recent decades, ionic liquids (ILs) have attracted a great deal of interest from the scientific community as a result of their special physicochemical characteristics [1]. These molten salts at temperatures below 100 °C are characterized by very low vapor pressure, high chemical and thermal stability, high solubilization ability and good electrical conductivity. These peculiarities alone would be enough to make IL materials of great potential in numerous applications; however, it is the possibility to combine up to 1 trillion cations and anions that makes them extraordinarily versatile [2]. Organic cations combined with anions of organic or inorganic nature produce salts with very different properties, adaptable for a given application.

According to their ability to donate a proton, ILs have been classified into aprotic ILs (AILs) and protic ILs (PILs), the latter having a variable proton activity of great influence in chemical and biological processes [3]. Both classes of salts have been initially studied with the main purpose of using them as green solvents in substitution of organic solvents. However, in the last decade, the investigation of new uses for the known ILs, the synthesis and characterization of new ones, as well as toxicity studies [4], have opened new horizons regarding their applicability in many fields of science and technology [5]. This fact is reflected in the academic papers published on applications of ILs in the last decade [6], observing that the interest in electrochemical (supercapacitors, batteries, electrodeposition or sensors) and industrial (separation, extraction or dissolution) applications is maintained and, despite being the majority, those related to catalysis are reduced. At the same time, there has been a significant increase in the number of areas that has started using advanced IL-based materials.

Special attention needs to be paid to the emerging and promising fields of ILs in biotechnology and bioengineering. Since the first papers on biotransformation were published in 2000, ILs have become not only a promise, but a reality in these areas. The possibility of availing materials with physical properties tailored to a specific application has been key for using them in other applications, such as stabilization and activation of biocatalysts, biomass treatment processes, synthesis of biopolymer-based hydrogels, bioseparation processes, environmental problem solving or the development of new ILs from natural products [7]. More specifically, in biomedicine and the pharmaceutical industry, important advances have also been made in the formulation of advanced materials for the generation of artificial tissues [8], drug transport or the formulation of active pharmaceutical ingredients [9,10], enzyme stabilization [11], biocides [12], etc. These applications are only a small sample of the potential of these materials and the importance of research in future applications. As for the environmental field, materials capable of eliminating insecticides [13], filtering and purifying water [14], capturing carbon dioxide [15,16,17] or recovering metals [18] have been developed.

All these new advances and those that will be produced in the future are the result of basic science work, in which physicochemical characterization plays an important role. Thus, in the last 15 years, our research group has focused on the study of these materials contributing to their development from the perspective of physicochemical characterization [19,20,21,22]. More recently, we have worked with PILs of the alkylammonium nitrate family [23,24], interest in which by the scientific community has been growing steadily due to the promising applications developed in different fields of work. Among these PILs, ethylammonium nitrate (EAN) and propylammonium nitrate (PAN) deserve special mention, as they have been the subject of more than 700 bibliographic references up to 2020 [24], half of them devoted to exploring their potential applications. From then until now, numerous studies have been published demonstrating their potential [25,26,27,28,29,30,31,32,33,34,35,36,37,38,39,40].

The remaining half of these references are mainly devoted to physically and chemically characterizing these substances, reflecting a wide dispersion of results for the same magnitude at a given temperature. Much of the variation in these data appeared to be due to the presence of small amounts of water derived from the hygroscopic nature of these substances, as we showed in a previous paper [24]. There, we presented the correlation of electrical conductivity (κ), density (ρ), viscosity (ƞ), refractive index (n_D_) and surface tension (σ) with water concentration (w) below 30,000 ppm and temperature (T), or with both simultaneously for the first two properties. The results obtained make it possible to study the influence of other variables on the physical properties independently of the water content. This was the starting point of the present work.

The aim of this paper is to continue the study of the mixtures of EAN and PAN to achieve a deeper characterization of these materials. In this occasion, we measured the same physical properties of these two PILs doped with four inorganic salts of the same nitrate anion. These salts comprise monovalent (Li^+^), divalent (Ca^2+^ and Mg^2+^) and trivalent (Al^3+^) metal cations. The chosen concentration of the salt for all the mixtures was close to saturation in order to determine the effect of the maximum presence of added salt. Meanwhile, the system {EAN + LiNO_3_} was studied at several concentrations. Additionally, a study of the electrochemical potential windows (EPW) was also made for the most concentrated mixtures. This property, which has not been published previously for these mixtures, was determined at 25 °C by linear sweep voltammetry employing three different working electrodes (WEs): platinum, glassy carbon and graphite.

The ion radii (r_ion_) and the electrostatic field at the ion surface (Ur_ion_) of the incorporated cations are shown in Table 1. The final objective of this work is to know how the presence of these metal cations influences the properties mentioned above, and to provide a novel correlation between the physical properties of the EAN and w, T and lithium salt concentration ([Li^+^]), simultaneously.

## 2. Results

As mentioned above, in this work we present the continuation of the characterization of the aqueous mixtures of EAN and PAN [24], with the study of the physical and electrochemical properties of these PILs doped with several nitrate salts. Following this order, the physical properties of the PILs samples are first presented and then their electrochemical study follows.

### 2.1. Physical Properties of the PILs Samples

Mixtures of {EAN + LiNO_3_, Ca(NO_3_)_2_, Mg(NO_3_)_2_ or Al(NO_3_)_3_} and {PAN + LiNO_3_, Ca(NO_3_)_2_, Mg(NO_3_)_2_ or Al(NO_3_)_3_} were physically characterized. Additionally, in {EAN + LiNO_3_} several molal concentrations (m) of the metal cation were studied. m was defined in this work as mol of the incorporated cation per kg of IL.

Among the physical properties, κ, as well as ρ and ƞ, were measured when possible in a wide temperature range, between 5 and 95 °C. In contrast, n_D_ and σ were limited to 25 °C. EAN and PAN mixtures were, in general, liquids. However, some of them, even though they were liquids, presented too much viscosity to measure some of their properties; meanwhile, the studied mixtures with Ca(NO_3_)_2_ were solid at room temperature.

The experimental results obtained for the five physical properties of the EAN and PAN mixtures can be seen in Figure 1 and Figure 2, the experimental values of which are presented in Appendix A. The values without any water content, which were reported in our previous work [24], are also represented in these Figure 1 and Figure 2.

As has been explained, κ, ƞ and ρ were measured in a wide range of temperature. In this interval, PILs samples were in liquid state, except those with Ca(NO_3_)_2_. Among these solid mixtures, EAN + Ca(NO_3_)_2_ showed a very slow speed of physical change, so that measurements in liquid and solid state below 65 °C were possible, although there were little differences between both values, as can be seen in Figure 1a.

The experimental results in liquid samples for κ and ƞ were expressed as a function T in °C by the equation of Vogel–Tammann–Fulcher (VTF), as is usual in liquids of this nature [19,42,43,44,45]:(1)$$Q=A_{Q}e^{B_{Q}/\left(T+273.15-T_{0,\;Q}\right)},$$
where Q describes κ or ƞ, while A_Q_ is the limit of the corresponding property at infinite temperature, |B_Q_| is related to the activation energy of ion hopping or flow and, finally, T_0,Q_ is associated with the glass transition temperature in K.

On the other hand, density data showed a linear relationship described by Equation (2), where ρ_0_ is the value of the property at 0 °C and C the slope of the linear fit:(2)$$\rho =\rho_{0}+C\cdot T,$$

The fitting parameters for the physical properties are shown in Table 2 and Table 3, along with the composition of the samples based on the Karl Fischer (KF) titration for water and inductively coupled plasma mass spectroscopy (ICP-MS) analysis for the incorporated metal cation. w is presented in ppm, while the metal cation concentration ([Me^n+^]) is represented by the ICP-MS result in mg·g^−1^ and the calculated m. Additionally, the percentage deviation (s_%_), defined in Equation (3), of the VTF fittings are given, as well as the coefficient of determination (R^2^) of Equation (2).
(3)$$s_{\%}=100\;\sqrt{\frac{\sum_{i=1}^{N}{\left(\frac{Q_{\exp,\;i}-Q_{\mathrm{VTF},i}}{Q_{\exp,\;i}}\right)}^{2}}{N-1}}.$$

As n_D_ and σ were only measured at 25 °C, fitting parameters for them as functions of T were not calculated. Nevertheless, the composition of the corresponding samples coincided with that presented in Table 3 for those of ƞ and ρ.

### 2.2. Electrochemical Potential Windows of the PILs Samples

The EPWs at 25 °C for the PILs, EAN and PAN, and their mixtures {EAN + LiNO_3_ 2.01 m, Ca(NO_3_)_2_, Mg(NO_3_)_2_ or Al(NO_3_)_3_} and {PAN + LiNO_3_, Ca(NO_3_)_2_, Mg(NO_3_)_2_ or Al(NO_3_)_3_} were found by linear sweep voltammetry. Generally, the criterion of 5 mA·cm^−2^ was applied [46] or the potential (E) taken with the closest corresponding value to it.

Voltammograms were measured employing different three-electrode electrochemical cells symbolized by their WEs. Platinum (DPt) and glassy carbon (DGC) disks were used as WEs with a platinum counter electrode and a home-made Ag/${\mathrm{Ag}}_{\mathrm{IL}}^{+}$ reference electrode. The third cell was a screen-printed electrode (SPE) with a graphite WE and counter electrode and Ag as pseudo-reference electrode. Experiments with DPt and DGC as WEs were performed at 0.100, 0.010 and 0.001 V·s^−1^, while those with SPE were performed only at 0.100 V·s^−1^.

To obtain comparable results, we employed mixtures of the ILs with ferrocene (Fc) as standard. In this way, the electrochemical study of the standard redox couple ferrocene–ferrocenium (Fc-Fc^+^) let us escalate the voltammograms E and determine the effective area of the WEs (A_WE_) to convert intensity (I) into current density (J = I/A_WE_). The results for the Fc solutions are shown in Section 4.6.

As representative examples of the electrochemical study, the voltammograms measured at 0.100 V·s^−1^ are shown in Figure 3 and Figure 4, for EAN and PAN samples, respectively. These voltammograms and the remaining ones can be seen in Appendix A.

Regarding the electrochemical study, as was explained before, E was escalated so that the reduction formal potential (E^0′^) of Fc is established at 0 V. At the same time, this standard was employed to use J, correcting the different areas of the WEs. EPWs of PILs mixtures, determined at 0.100 V·s^−1^ by applying the defined criterion, are reported in Table 4. Data corresponding to EPWs of PILs samples for all the scan rates can be seen in Appendix A. However, as mentioned above, some limits were not established in the 5 mA·cm^−2^ criterion, being not possible to reach such value. The data reported under this condition are preceded in all the tables by > or <. Along with the EPWs, w and [Me^n+^] are presented in Table 4, except for w of those measured with SPE, since KF titrator was not available.

## 3. Discussion

### 3.1. Physical Properties as a Function of Temperature

κ, ƞ and ρ of the PILs and their mixtures {EAN + LiNO_3_, Ca(NO_3_)_2_, Mg(NO_3_)_2_ or Al(NO_3_)_3_} and {PAN + LiNO_3_, Ca(NO_3_)_2_, Mg(NO_3_)_2_ or Al(NO_3_)_3_} were studied in a range of temperatures, from 5 to 95 °C. Analyzing the results obtained, the usual behavior of liquids versus T was found. In this way, when T increased, κ did too, while ƞ and ρ decreased. Moreover, as shown in Table 2 and Table 3, κ and ƞ were fitted following a VTF equation due to their exponential trend, opposite to ρ that was linear. Finally, it should be highlighted that the properties related to mass transport, κ and ƞ, presented significant changes with respect to the temperature gradient, observing descends up to 97%, while in ρ was below 5% in all cases. This is an expected result, given that the increase in temperature favors the molecules’ mobility, which is precisely the most important factor in transport properties.

### 3.2. Physical Properties as a Function of Salt Doping

The main problem when trying to study the effect of salt doping, with LiNO_3_, Ca(NO_3_)_2_, Mg(NO_3_)_2_ or Al(NO_3_)_3_, on the physical properties (κ, ƞ, ρ, n_D_ and σ) of EAN and PAN was the diverse composition of the samples, as shown in Table 2 and Table 3. Therefore, the influence of water in each sample was corrected in order to study the salt doping effect alone. This influence was calculated by employing the information provided in our previous published study, which preceded this work [24]. In this study, we obtained the following equations for EAN and PAN that relate their κ in mS·cm^−1^ and ƞ in mPa·s with w in ppm and T in °C, as well as n_D_ and σ in mN·m^−1^ as functions of w in ppm at 25 °C:(4)$$\kappa_{\mathrm{EAN}}\left(w,\;T\right)=666.2e^{-424.9/\left(T+273.15-175.8\right)}+\left(1.554\cdot 10^{-4}+7.445\cdot 10^{-6}T-\;1.549\cdot 10^{-8}T^{2}\right)w,$$
(5)$${ƞ}_{\mathrm{EAN}}\left(w,\;T\right)=0.2111e^{778.8/\left(T+273.15-148.7\right)}-e^{-6.246-4.994\cdot 10^{-2}T+1.740\cdot 10^{-4}T^{2}}w,$$
(6)$$n_{D,\;\mathrm{EAN}}\left(w,\;25{}^{\circ}C\right)=1.45395-5.14135\cdot 10^{-8}w-3.00678\cdot 10^{-12}w^{2},$$
(7)$$\sigma_{\mathrm{EAN}}\left(w,25{}^{\circ}C\right)=47.75+2.008\cdot 10^{-5}w,$$
(8)$$\kappa_{\mathrm{PAN}}\left(w,\;T\right)=904.1e^{-620.8/\left(T+273.15-166.4\right)}+\left(6.294\cdot 10^{-5}+5.924\cdot 10^{-6}T\;-1.775\cdot 10^{-8}T^{2}\right)w,$$
(9)$${ƞ}_{\mathrm{PAN}}\left(w,\;T\right)=0.1823e^{869.5/\left(T+273.15-154.1\right)}-e^{-4.917-5.883\cdot 10^{-2}T+1.951\cdot 10^{-4}T^{2}}w\;+e^{-16.22-6.074\cdot 10^{-2}T+1.633\cdot 10^{-4}T^{2}}w^{2},$$
(10)$$n_{D,\;\mathrm{PAN}}\left(w,\;25{}^{\circ}C\right)=1.45536-1.08059\cdot 10^{-7}w,$$
(11)$$\sigma_{\mathrm{PAN}}\left(w,25{}^{\circ}C\right)=38.72+2.201\cdot 10^{-5}w,$$

Additionally, from the data reported in that work, we now present the corresponding equations that define ρ of EAN and PAN as functions of w in ppm and T in °C:(12)$$\rho_{\mathrm{EAN}}\left(w,\;T\right)=1.2257-5.9379\cdot 10^{-4}T\;\;+\left(-1.4311\cdot 10^{-7}-3.2075\cdot 10^{-10}T-2.5626\cdot 10^{-12}T^{2}\right)w,$$
(13)$$\rho_{\mathrm{PAN}}\left(w,\;T\right)=1.1664-5.8986\cdot 10^{-4}T\;\;+\left(-6.2178\cdot 10^{-8}-5.9006\cdot 10^{-10}T\right)w,$$

Therefore, the doping effect was the difference of the properties between the mixtures (Q_mix_) and pure PILs (Q_PIL_), ∆Q = Q_mix_ − Q_PIL_, all extrapolated to null content of water by using Equations (4)–(13). At the same time, the data obtained were normalized for comparing purposes dividing by m of the respective metal cation added, which are shown in Table 2 and Table 3.

The final calculated effects are graphically represented in Figure 5 for EAN, and Figure 6 for PAN. All data are directly presented, apart from σ results in Figure 5e and Figure 6e. In these two graphics, the absolute values are presented, where open dots imply a decreasing effect on the property and solid dots an increasing one.

As can be observed, the addition of the different salts generated a similar effect on the physical properties of both PILs, EAN and PAN. The type of influence, as well as its significance, were related to the structural changes made by the incorporation of the metal cations. In Table 5, the molal percentage influences observed for the maximum [Me^n+^] studied on each physical property and PIL, at the extremes of the temperature range and 25 °C, are reported. As can be seen in this table, the absolute molal percentage influences are reduced when temperature rises. Moreover, the two mass transport properties, κ and ƞ, showed again the larger values, which were associated to their susceptibility to structural changes that affect the ion mobility.

Initially, EAN and PAN have similar bulk structures that can mainly be described by the existence of two types of domains: the polar one, consisted of the nitrate anions and the polar heads of the cation, and the apolar domain, formed by the aggregation of the alkyl chains [47,48,49,50,51,52,53,54,55]. In previous works, the influence of small quantities of water was reported, which is incorporated in polar domains playing the role of a lubricant, increasing the mobility of IL ions [24,56,57,58,59,60,61]. Following the same way, the new metal cations are also added to the polar domain [41,62,63,64,65,66,67,68,69,70]. Nevertheless, their interactions inside go the opposite way.

The metal cations tend to place themselves in the structural holes of the polar domains, where they compete against the alkylammonium cations to coordinate with the nitrate anions. This coordination depends on the cation characteristics [41], of which r_ion_ and Ur_ion_ are shown in Table 1. Recent works published show that the lithium cation mainly presents a tetrahedral coordination with four nitrates anions, while the others (Ca^2+^, Mg^2+^ and Al^3+^) prefer an octahedral one with six [41,62].

Although, globally, the structure does not present significant changes from the pure PILs, at a nanoscale level the competition between cations weakens and breaks H-bonds interactions while it increases their total number [41,64,65,66]. Moreover, the coordinated metal cations form a kind of clusters, which give way to new networks that increase the ordination and show characteristics close to those of solid crystal structures [64,65,67,68,69].

Consequently, two effects relevant to our discussion take place: the creation of clusters and networks and the reduction in structural holes. Considering these two factors, the effects observed in most of the physical properties studied could be explained. Firstly, the formation of these networks reduces the mobility of ions, meaning that ƞ of the mixture increases and κ decreases as consequence. Secondly, the reduction in total structural holes and the increase in comprising, due to the clusters formed, explains the increase in ρ and n_D_, the last one being due to the decrease in places where the light can pass without deviation.

The magnitude of the different effects can also be associated to the structural changes. As the metal cation valence increase, stronger interactions are established and, consequently, alterations that are more significant happen [41,62,63]. This is especially visible with the lithium cation, the lesser valence of which only lets it coordinate with a lower number of anions; therefore, a weaker influence is produced [62,63]. Comparing among the metal cations, the order of their effect per molal unit can be generally related to the Ur_ion_ of the metal cations that were reported in Table 1. In this way, the higher this characteristic is, the more significant is the influence of the incorporated metal cation. The only exceptions observed comes from the PAN + Ca(NO_3_)_2_ mixture, which is in solid state, more substantially limiting the ion movement, and in ƞ, where the great difference between r_ion_ of Mg^2+^, with the largest size, and Al^3+^, the smallest, makes the trivalent cation show an inferior variation.

Surface tension is a special case for discussion. This property depends not only on the ions in the sample, but also on the interactions of the surface ions with the atmosphere. In fact, a stratification of the structure is found in these PILs, transiting from the bulk to the surface [51,71,72,73,74].

As stated above, the addition of the metal cations in the bulk structure increases the structure ordination at a nanoscale level. When these cations may be situated close to the surface layer, their influence may let, indirectly, the surface structure to gain some ordination, so that σ increases. However, contrary to the general trend, the aluminum cation generates a decrease in this property. This influence could be associated to having the highest valence while also the smallest size, which should produce a greater alteration of the original bulk structure, as explained above. With respect to σ, aluminum cations may be reaching the surface layer thanks to their small size, where, due to their great Ur_ion,_ the structure breaks, thus observing a fall of the property. Lithium cations also reach the surface layer at higher contents, as will be explained below, but opposite to Al^3+^ they induce the nanostructure ordination [69]. Despite these differences, the amount of variation seen in Figure 5e and Figure 6e continues to follow the discussion previously presented, depending on the metal cations’ Ur_ion_.

Having analyzed the different incorporated salts, we will now focus on the EAN + LiNO_3_ mixtures, in which several [Li^+^] were studied. The calculated data for the physical properties of these samples, correcting the influence of the water presence as described at the start of this section, are showed in Figure 7.

The results in Figure 7 show that the properties varied with the incorporated [Li^+^], although the effect of the salt per molal unit was reduced as the concentration increased. This was associated to the major change in the structure happening with a low content of the lithium salt, that weakens the H-bonds widening the structural holes where the next lithium cations are better situated [63].

The trend in the properties, except for σ, can be described by a polynomial equation such as:(14)$$Q=Q_{\mathrm{IL}}+\sum_{i=1}^{N}A_{i}{\left[{\mathrm{Li}}^{+}\right]}^{i},$$
where Q is the physical property, Q_IL_ the value of the property for the pure IL and A_i_ the fitting parameters. The values of Q_IL_ and A_i_, together with the coefficient of determination R^2^, are shown in Appendix A.

On the other hand, in Figure 7e, two jumps were appreciated for σ: one when salt is added and another between 1.29 and 1.59 m. These jumps fitted with the work of Hjalmarsson et al. [69], in which they explain two structural changes in the mixtures of EAN with LiNO_3_. The first change is when the salt is added, altering the bulk structure as explained above, which also produces a certain level of ordination in the surface layer. Nevertheless, the lithium cations cannot reach the surface. It is when the content reaches 10% *w*/*w* (1.59 m) that the metal cations can occupy this layer, thus significantly altering the surface layer again.

In our previous work, we defined equations to express κ_EAN_ and ƞ_EAN_ of the aqueous mixtures of EAN as functions of w and T [24]. In this work, we pretend to widen the functionality of Equations (4)–(6) and (12) by including the effect of lithium nitrate doping. Therefore, factors dependent on [Li^+^] in m and T in °C were developed. We finally reached the following equations:(15)$$\kappa_{\mathrm{EAN}}\left(w,\;T,\;\left[{\mathrm{Li}}^{+}\right]\right)=666.2e^{-424.9/T+273.15-175.8}+\left(1.554\cdot 10^{-4}+7.445\cdot \;10^{-6}T-1.549\cdot 10^{-8}T^{2}\right)w+\left(-2.785-0.1178T\right)\left[{\mathrm{Li}}^{+}\right]+\left(0.4671+8.991\cdot \;10^{-3}T+3.582\cdot 10^{-4}T^{2}\right){\left[{\mathrm{Li}}^{+}\right]}^{2},$$
(16)$$\rho_{\mathrm{EAN}}\left(w,\;\left[{\mathrm{Li}}^{+}\right],\;T\right)=1.2257-5.9379\cdot 10^{-4}T\;+\left(-1.4311\cdot 10^{-7}-3.2075\cdot \;10^{-10}T-2.5626\cdot 10^{-12}T^{2}\right)w+\left(4.146\cdot 10^{-2}-1.368\cdot 10^{-4}T+1.191\cdot \;10^{-6}T^{2}\right)\left[{\mathrm{Li}}^{+}\right]+\left(-6.354\cdot 10^{-3}+6.063\cdot 10^{-5}T-5.792\cdot 10^{-7}T^{2}\right){\left[{\mathrm{Li}}^{+}\right]}^{2},$$
(17)$${ƞ}_{\mathrm{EAN}}\left(w,\;T,\;\left[{\mathrm{Li}}^{+}\right]\right)=0.2111{*e}^{778.8/T+273.15-148.7}-\;e^{-6.246-4.994\cdot 10^{-2}T+1.740\cdot 10^{-4}T^{2}}w+e^{4.136-5.197\cdot 10^{-2}T+1.777\cdot 10^{-4}T^{2}}\left[{\mathrm{Li}}^{+}\right]-\;e^{1.857-2.975\cdot 10^{-2}T}{\left[{\mathrm{Li}}^{+}\right]}^{2},$$
(18)$$n_{D,\;\mathrm{EAN}}\left(w,\;\left[{\mathrm{Li}}^{+}\right]\right)=1.45395-5.14135\cdot 10^{-8}w-3.00678\cdot 10^{-12}w^{2}+\;3.06575\cdot 10^{-3}\left[{\mathrm{Li}}^{+}\right],$$
where s_%_ when recalculating the experimental results were 1.2, 0.02, 1.3 and 0.014, respectively.

Few works about the physical properties of these systems are found out of our group or collaborations. ƞ results of the system EAN + LiNO_3_ were compared with those published by Hjalmarsson et al. and Prabhu et al. [69,70]. The data presented in this work fit well with those published, as they fall in the middle of them, being a bit lower than the data reported by Hjalmarsson et al. [69] and slightly higher than those of Prabhu et al. [70]. 

### 3.3. Electrochemical Study

In the previous Figure 3 and Figure 4, we showed the voltammograms at 0.100 V·s^−1^ of the PILs, and their highest concentrated salt mixtures, measured with DPt, DGC and SPE as WEs. In these voltammograms, it can be observed that the EPW changed in greater proportion in the cathodic limit, associated to the reduction of protons to hydrogen [75,76,77,78], which is affected by the present cations in each different sample, while the anodic one was limited by oxidation reactions of the ILs [77,78]. Exception are the mixtures of PAN with calcium or magnesium salts, in which their low κ (showed in Figure 2 and Appendix A) may be influencing the EPW. Especially PAN + Mg(NO_3_)_2_, in which the sample generates a minimal I response to the application of E, generating an extreme enlargement at both limits of the EPW.

The EPWs at 0.100 V·s^−1^ of the PILs mixtures were reported in Table 4. In this table, a general trend can be observed, so that the EPW followed the order PIL + Al(NO_3_)_3_ < PIL < PIL + LiNO_3_ < PIL + Ca(NO_3_)_2_. The sole exception at 0.100 V·s^−1^ is EAN measured in SPE, which EPW is only below its mixture with the calcium salt. For its part, PIL + Mg(NO_3_)_2_ showed variable behavior. EAN + Mg(NO_3_)_2_ has a EPW between EAN + LiNO_3_ and EAN + Ca(NO_3_)_2_ with SPE and DPt, while with DGC it presented a EPW no significantly narrower than EAN + LiNO_3_. Meanwhile, as stated above, the low I respond of PAN + Mg(NO_3_)_2_ may be widening its EPW, becoming the largest without getting close to the criterion and only being possible to measure at this scan rate.

The mentioned order at 0.100 V·s^−1^ can be often extrapolated to the other scan rates applied, which are reported in Appendix A. The alterations from this order happened with EAN + Mg(NO_3_)_2_ at 0.001 V·s^−1^, in which the criterion was not reached, and with PAN + Al(NO_3_)_3_, where the peak that established the cathodic limit changed, as is visible in Appendix A.

The order obtained fits with the r_ion_ of the different incorporated metal cations, where the addition of Al(NO_3_)_3_ generally produced a great narrowing of the EPW of both PILs. This reduction in the property was associated to the effect of Al^3+^ in the bulk structure due to both its smallest size and greatest Ur_ion_, as explained before, thus producing a similar effect that the one observed in σ.

## 4. Materials and Methods

### 4.1. Materials and Sample Preparation

Mixtures of two PILs, EAN and PAN, with the inorganic nitrate salts of lithium, calcium, magnesium and aluminum have been studied. Specifically in the mixture of EAN with LiNO_3_, several mixtures of different concentrations have been prepared.

All samples were prepared from the compounds described in Table 6. Prior to their use, the commercial ionic liquids have been dried at a pressure of 10 mbar and at 120 °C for 10 h, mixing by weight trying to reach saturation under a controlled humidity atmosphere (<15%) and homogenizing by stirring and hot ultrasonic bath (80 °C). As water is added with the salts, the mixtures are again subjected to the same drying procedure, being the final concentrations of water and metal cation determined prior to their measurement, by Karl Fischer titration techniques and ICP-MS, respectively. The sample for KF titration were taken under the controlled atmosphere described above and measured with a Karl Fischer C10S coulometric titrator, from Mettler Toledo. Uncertainties of Karl Fischer titrations depends on the sample mass, measured with a Baxtran HZ 2201 balance with an uncertainty of 1 × 10^−4^ g, and the titrator, so they are given in each w. About ICP-MS, measurements were made by the Research Support Services of the University of A Coruña.

### 4.2. Electrical Conductivity

The electrical conductivity of the samples with protic ionic liquids was measured between 5 and 95 °C using a GLP 31 Crison conductivity meter, which operates at 500 Hz AC current and 0.5 V voltage. Coupled to it, a 52 92 Hach measuring cell was used, which allows measurement between −30 and 80 °C with an associated error of 0.5%. The stability of the physical property in the samples allows the use of this cell for higher temperatures. Temperature control was carried out with a Julabo F25 thermostatic bath with an uncertainty of 0.1 °C.

### 4.3. Density and Viscosity

The density and viscosity of the samples with protic ionic liquids were measured simultaneously with an Anton Paar Stabinger VTM 3000 viscometer between 5 and 95 °C. This equipment provides values with uncertainties of 0.5 kg·m^−3^ and 0.4% of the viscosity. The temperature control was carried out by the Peltier effect with an uncertainty of 0.02 °C. The equipment used allows the measurement of fluids with a maximum viscosity of 20 Pa·s^−1^, so no measurements have been made of solid or excessively viscous mixtures, for which values for refractive index and surface tension are also not shown.

### 4.4. Refractive Index

The refractive index of samples with protic ionic liquids was measured at 25 °C with an Anton Paar Abbemat-WR refractometer, which provides measurements with an uncertainty of 1 × 10^−5^ and has an internal Peltier thermostatization system, with an uncertainty of less than 0.03 °C, for temperature control.

### 4.5. Surface Tension

The surface tension of the samples with protic ionic liquids was measured at 25 °C with a Lauda TVT1 automatic tensiometer, which provides measurements with an uncertainty of less than 0.02 mN·m^−1^. The needle with the sample has a jacket connected to a Lauda RC6 CP thermostatic bath, which ensures an uncertainty better than 0.2 °C.

### 4.6. Electrochemical Potential Windows

The electrochemical potential windows were determined at 25 °C by measuring the voltammograms of the different ILs and their mixtures, setting the limits at 5 mA·cm^−2^ [46], at the physical evidence of chemical reactions (bubbles) or at the potential with the nearest response when having applied the maximum potential range. The voltammetric curves were determined by linear sweep voltammetry using different three-electrode electrochemical cells.

On the one hand, a DRP C110 screen-printed electrode (SPE) from DropSens was used, connected to an Autolab PGSTAT302 potentiostat from Metrohm. This SPE contains a three-electrode electrochemical cell consisting of a 4 mm diameter graphite working electrode, a graphite auxiliary electrode and a silver pseudo-reference electrode. The measurements with this system were made at 0.100 V·s^−1^ from −2 to 2 V and are open to the atmosphere, controlling the temperature by the laboratory air conditioning.

On the other hand, two working disk electrodes of 3 mm diameter from Metrohm, one of platinum (DPt) and the other of glassy carbon (DGC), which were connected to an Autolab PGSTAT204 potentiostat from Metrohm, were used. For both disk electrodes, electrochemical cells were configured with a 1 cm^2^ platinum auxiliary electrode from Metrohm and a home-made Ag/${\mathrm{Ag}}_{\mathrm{IL}}^{+}$ reference electrode. This type of home-made electrode is often used with ILs, as can be seen in other published works with electrochemical studies of EAN and PAN [75,79,80]. This electrode was made in situ and consisted of a silver wire immersed in a saturated solution of silver nitrate in the corresponding IL, placing a bridge composed of the same IL between the electrode and the sample to avoid cross pollution. Measurements with this system start from the base potential of the sample towards the extremes at 0.100, 0.010 and 0.001 V·s^−1^. The same sample aliquots were studied with both WEs, generally with DPt first and DGC after. The temperature was controlled by a thermal jacket connected to a PolyScience 9101 thermostatic bath, which ensures an uncertainty of 0.1 °C, and an argon atmosphere was employed, an inert gas with which the samples are bubbled prior to their measurement. Likewise, prior to their use, the working electrodes are polished using the procedure present in the literature [81].

Both potentiostats have an accuracy of 0.2% for potential and 0.2% for intensity. Ferrocene was used as reference for all the electrochemical cells studied, scaling the voltammetric curves to a formal reduction potential (E’^0^) of the redox pair Fc-Fc^+^ of 0 V. In Table 7, the E^’0^ obtained is reported, together with the effective areas of WE (A_WE_). These areas were employed to convert the signal response, when applying the potential, from intensity (I) to current density (J = I/A_WE_). A_WE_ values were determined experimentally with the electrochemical study of the redox couple Fc-Fc^+^ in EAN and PAN, in which the diffusion coefficients of ferrocene (Fc) are published by Shotwell et al. [79].

## 5. Conclusions

To sum up, physical and electrochemical characterizations were made of the PILs, EAN and PAN, doped with four nitrate salts which added monovalent (Li^+^), divalent (Ca^2+^ and Mg^2+^) or trivalent (Al^3+^) metal cations.

The results obtained were analyzed as functions of the temperature, when several temperatures were tested, and the salt incorporated. κ, ƞ and ρ vary versus temperature like expected, increasing κ while decreasing ƞ and ρ at higher temperatures. Moreover, this variation is far more significant in the mass transport properties, detecting within the range studied (5–95 °C) falls up to 97%, while in ρ it was in all cases below 5%.

In relation to the metal cations added, their influence on the physical properties was associated to the structural changes that they generate when entering the polar domains of the PILs. At the same time, the magnitude of their effects, the molal percentage value of which descends with temperature, were generally linked to their Ur_ion_, although the influence of other characteristics such as metal cation size and the solid physical state of the PAN + Ca(NO_3_)_2_ mixture were also considered. In this way, at 25 °C, the addition of Li^+^ to EAN presented the less significant reduction in κ (−21.15%/m), followed by Ca^2+^ (−53.20%/m), Mg^2+^ (−58.98%/m) and Al^3+^ (−74.26%/m). The absolute molal percentage value descends with temperature in the mass transport properties, for example the effect in the κ of EAN + LiNO_3_ 1.59 m mixture varies from −25.76%/m at 5 °C to −9.91%/m at 95 °C.

Specifically, in the mixtures of EAN with LiNO_3_, in which several contents of Li^+^ were studied, an increase in the effect is seen as concentrations rise. However, a reduction in its influence per molal unit was also detected, being associated to interactions between the metal cations. The study of this system, along with that of a previous work, allowed us to develop novel predictive equations for most of the physical properties presented as functions of up to three variables: w, T and [Li^+^].

Electrochemical studies showed varied results where the effects on the EPWs of PIL mixtures was generally greater in the cathodic limit. Usually, EPW followed a sequential order that fitted with the r_ion_ of the incorporated metal cations, where the addition of Al(NO_3_)_3_ narrowed the EPWs of both PILs. However, exceptions in this order were found to be related to the solid state of mixtures with Ca(NO_3_)_2_, the physical properties of PAN + Mg(NO_3_)_2_ or changes in the scan rate applied.

## Figures and Tables

**Figure 1 ijms-23-16040-f001:**
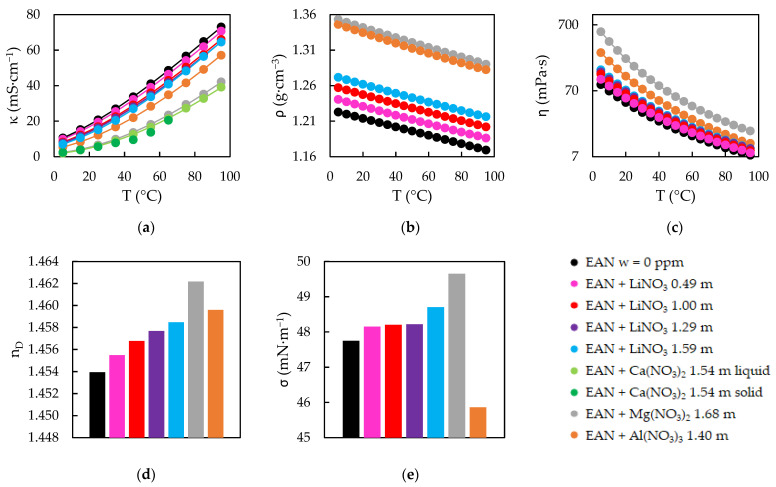
Experimental electrical conductivities, κ (**a**), densities, ρ (**b**), viscosities, ƞ (**c**), refractive indexes, n_D_ (**d**) and surface tensions, σ (**e**) of EAN at w = 0 ppm [24] and its mixtures. Solid lines were obtained from the corresponding fitting Equations (1) and (2), except the black ones.

**Figure 2 ijms-23-16040-f002:**
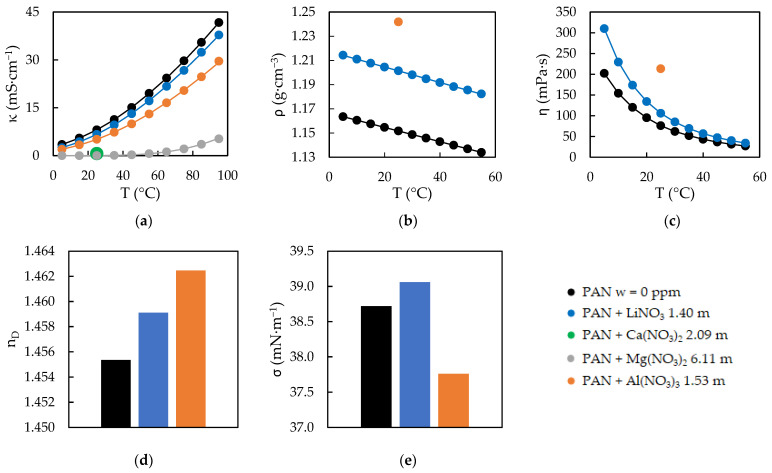
Experimental electrical conductivities, κ (**a**), densities, ρ (**b**), viscosities, ƞ (**c**), refractive index, n_D_ (**d**) and surface tensions, σ (**e**) of PAN at w = 0 ppm [24] and its mixtures. Solid lines were obtained from the corresponding fitting Equations (1) and (2), except the black ones.

**Figure 3 ijms-23-16040-f003:**
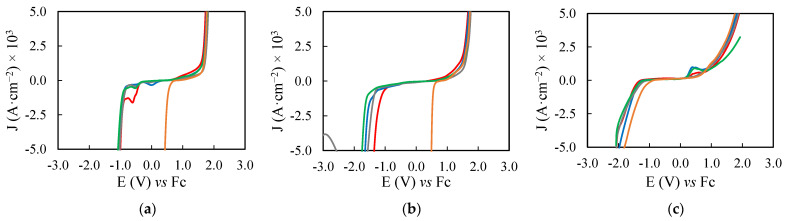
Voltammograms, at 0.100 V·s^−1^ with DPt (**a**), DGC (**b**) and SPE (**c**) as WEs, of EAN (**—**) and its mixtures with LiNO_3_ 2.01 m (**—**), Ca(NO_3_)_2_ 1.54 m (**—**), Mg(NO_3_)_2_ 1.68 m (**—**) or Al(NO_3_)_3_ 1.40 m (**—**).

**Figure 4 ijms-23-16040-f004:**
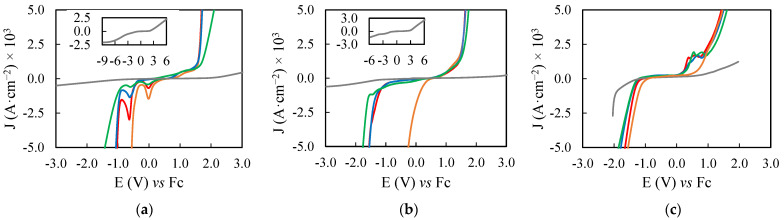
Voltammograms, at 0.100 V·s^−1^ with DPt (**a**), DGC (**b**) and SPE (**c**) as WEs, of PAN (**—**) and its mixtures with LiNO_3_ 1.40 m (**—**), Ca(NO_3_)_2_ 2.09 m (**—**), Mg(NO_3_)_2_ 6.11 m (**—**) or Al(NO_3_)_3_ 1.53 m (**—**). PAN + Mg(NO_3_)_2_ voltammograms are fully shown in snapshot.

**Figure 5 ijms-23-16040-f005:**
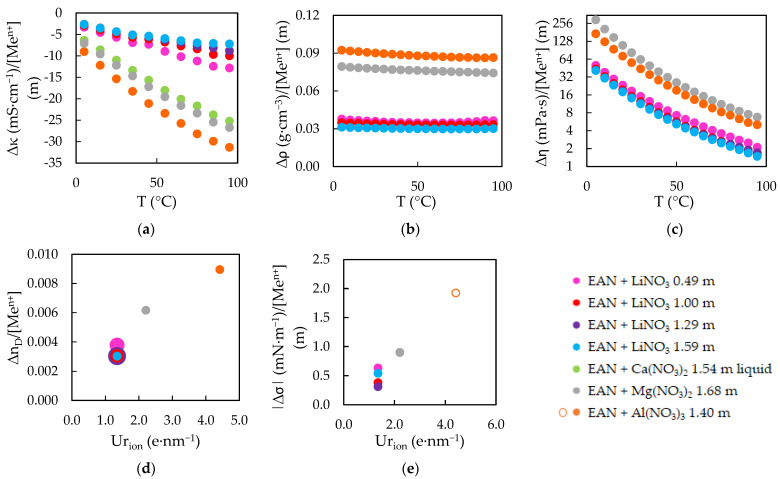
Doping effect per molal unit (∆Q/[Me^n+^]) in electrical conductivity, κ (**a**), density, ρ (**b**), viscosity, ƞ (**c**), refractive index, n_D_ (**d**) and surface tension, σ (**e**), of EAN mixtures. In Figure 5e, the absolute values of the effect are represented, with increasing (solid dot) and decreasing (open dot) effects occurring.

**Figure 6 ijms-23-16040-f006:**
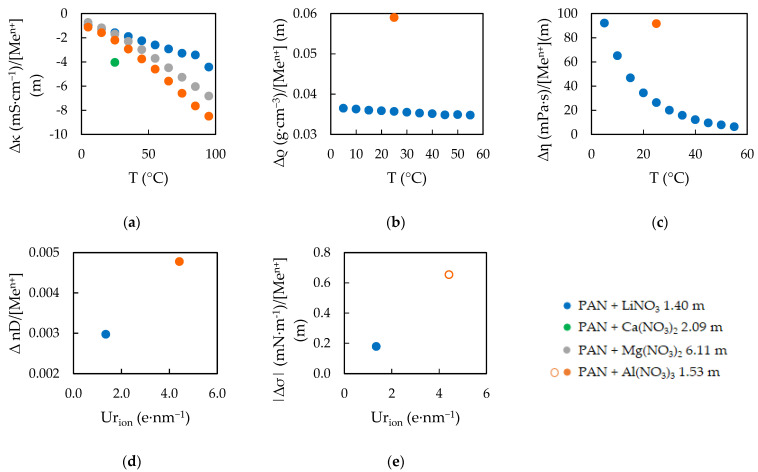
Doping effect per molal unit (∆Q/[Me^n+^]) in electrical conductivity, κ (**a**), density, ρ (**b**), viscosity, ƞ (**c**), refractive index, n_D_ (**d**) and surface tension, σ (**e**), of PAN mixtures. In Figure 6e, the absolute values of the effect are represented, with increasing (solid dot) and decreasing (open dot) effects occurring.

**Figure 7 ijms-23-16040-f007:**
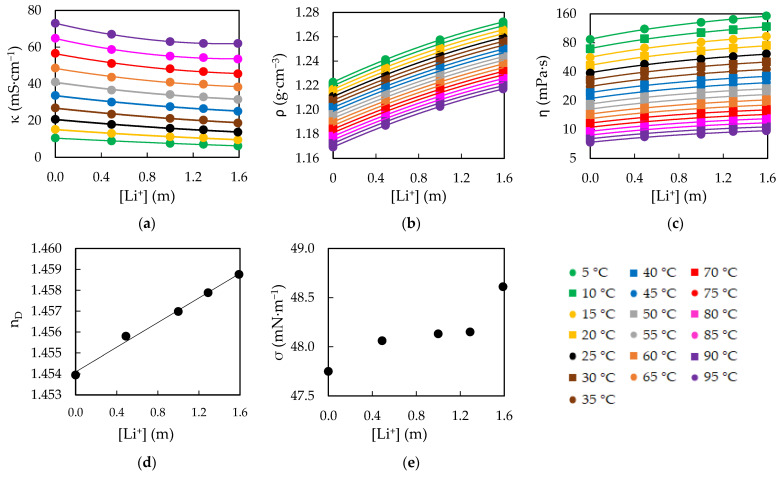
Calculated data at null water content for electrical conductivity, κ (**a**), density, ρ (**b**), viscosity, ƞ (**c**), refractive index, n_D_ (**d**) and surface tension, σ (**e**), of EAN mixtures with LiNO_3_ at several temperatures. Solid lines were obtained from the corresponding fitting Equation (14).

**Table 1 ijms-23-16040-t001:** Effective ionic radii (r_ion_) and the electrostatic field at the ion surface (Ur_ion_) of the incorporated salt cations [41].

Salt Cation	r_ion_ (Å)	Ur_ion_ (e·nm^−1^)
Li^+^	0.59	1.35
Ca^2+^	1.0	1.59
Mg^2+^	0.72	2.21
Al^3+^	0.54	4.42

**Table 2 ijms-23-16040-t002:** Fitting of electrical conductivity (mS·cm^−1^) experimental data of EAN and PAN mixtures as function of temperature: water (w) and doped cation ([Me^n+^]) content, parameters of Equation (1) and percentage deviation (s_%_).

Sample	w (ppm)	[Me^n+^] (mg·g^−1^)	[Me^n+^] (m)	A_κ_ (mS·cm^−1^)	B_κ_ (K)	T_0,κ_ (K)	s_%_
EAN + LiNO_3_	4730 ± 50	3.3 ± 0.1	0.49	664.4	−432.4	175.8	0.4
EAN + LiNO_3_	3430 ± 50	6.5 ± 0.3	1.00	649.2	−430.5	179.9	0.7
EAN + LiNO_3_	3270 ± 50	8.2 ± 0.4	1.29	680.8	−445.5	178.9	0.5
EAN + LiNO_3_	4300 ± 50	9.9 ± 0.6	1.59	782.0	−474.7	177.6	0.7
EAN + Ca(NO_3_)_2_	6600 ± 120	49 ± 1	1.54	700.4	−512.1	190.9	0.8
EAN + Mg(NO_3_)_2_	19,320 ± 60	32 ± 1	1.68	783.6	−529.5	187.3	1.0
EAN + Al(NO_3_)_3_	38,870 ± 70	28 ± 1	1.40	914.6	−541.0	173.0	0.5
PAN + LiNO_3_	3890 ± 50	8.8 ± 0.1	1.40	614.7	−510.0	185.4	0.8
PAN + Ca(NO_3_)_2_	4870 ± 50	62 ± 1	2.09	-	-	-	-
PAN + Mg(NO_3_)_2_	11,350 ± 70	77 ± 1	6.11	2393	−911.5	218.4	3
PAN + Al(NO_3_)_3_	1960 ± 50	31 ± 2	1.53	607.5	−581.2	176.6	1.0

**Table 3 ijms-23-16040-t003:** Fitting of density (g·cm^−3^) and viscosity (mPa·s) experimental data of EAN and PAN mixtures as function of temperature: water (w) and doped cation ([Me^n+^]) content, parameters and percentage deviation (s_%_) of Equation (1) as well as parameters and coefficient of determination (R^2^) of Equation (2).

Sample	w (ppm)	[Me^n+^] (mg·g^−1^)	[Me^n+^] (mol·kg_IL_^−1^)	A_ƞ_ (mPa·s)	B_ƞ_ (K)	T_0, ƞ_ (K)	s_%_	ρ_0_ (g·cm^−3^)	C × 10^4^ (g·cm^−3^·°C^−1^)	R^2^
EAN + LiNO_3_	4730 ± 50	3.3 ± 0.1	0.49	0.2116	793.4	150.1	0.2	1.243	−6.029	0.9994
EAN + LiNO_3_	3430 ± 50	6.5 ± 0.3	1.00	0.2094	803.5	152.4	0.2	1.260	−6.124	0.9997
EAN + LiNO_3_	3270 ± 50	8.2 ± 0.4	1.29	0.2645	743.9	158.9	0.2			
EAN + LiNO_3_	4300 ± 50	9.9 ± 0.6	1.59	0.2311	785.0	156.3	0.2	1.274	−6.173	0.9997
EAN + Mg(NO_3_)_2_	19,320 ± 60	32 ± 1	1.68	0.3094	775.1	174.6	0.3	1.356	−6.945	0.9998
EAN + Al(NO_3_)_3_	40,050 ± 70	28 ± 1	1.40	0.1548	898.3	157.5	0.4	1.349	−7.113	0.9991
PAN + LiNO_3_	3890 ± 50	8.8 ± 0.1	1.40	0.1740	893.3	158.8	0.3	1.217	−6.407	0.9998
PAN + Al(NO_3_)_3_	1960 ± 50	31 ± 2	1.53	-	-	-	-	-	-	-

**Table 4 ijms-23-16040-t004:** Electrochemical potential windows (V) of EAN and PAN mixtures with each working electrode (WE) at 0.100 V·s^−1^ as scan rate (ν): cathodic and anodic limits and water (w) and salt ([Me^n+^]) content.

Sample	w (ppm)	[Me^n+^] (mg·g^−1^)	[Me^n+^] (mol·kg_IL_^−1^)	WE	Cathodic Limit (V)	Anodic Limit (V)	EPW (V)
EAN	2380 ± 50	-	-	DPt	−0.998	1.732	2.730
EAN + LiNO_3_	2800 ± 50	12.2 ± 0.4	2.01	DPt	−1.019	1.761	2.780
EAN + Ca(NO_3_)_2_	6600 ± 120	49 ± 1	1.54	DPt	−1.074	1.783	2.857
EAN + Mg(NO_3_)_2_	19,320 ± 60	32 ± 1	1.68	DPt	−1.024	1.821	2.845
EAN + Al(NO_3_)_3_	38,870 ± 70	28 ± 1	1.40	DPt	0.429	1.774	1.345
EAN	5420 ± 50	-	-	DGC	−1.357	1.670	3.027
EAN + LiNO_3_	5590 ± 50	12.2 ± 0.4	2.01	DGC	−1.645	1.694	3.339
EAN + Ca(NO_3_)_2_	12,970 ± 60	49 ± 1	1.54	DGC	−1.746	1.734	3.480
EAN + Mg(NO_3_)_2_	25,820 ± 70	32 ± 1	1.68	DGC	−1.560	1.764	3.324
EAN + Al(NO_3_)_3_	41,350 ± 110	28 ± 1	1.40	DGC	0.493	1.732	1.239
EAN	-	-	-	SPE	−2.060	1.900	3.960
EAN + LiNO_3_	-	10.8 ± 0.2	1.75	SPE	−1.972	1.826	3.798
EAN + Ca(NO_3_)_2_	-	57.9 ± 1.0	1.91	SPE	<−2.075	>1.929	>4.004
EAN + Mg(NO_3_)_2_	-	33.1 ± 0.2	1.72	SPE	−2.065	1.880	3.945
EAN + Al(NO_3_)_3_	-	21.5 ± 0.8	0.98	SPE	−1.806	1.758	3.564
PAN	1300 ± 50	-	-	DPt	−1.031	1.694	2.725
PAN + LiNO_3_	3890 ± 50	8.8 ± 0.1	1.40	DPt	−1.066	1.714	2.780
PAN + Ca(NO_3_)_2_	4870 ± 50	62 ± 1	2.09	DPt	−1.424	2.095	3.519
PAN + Mg(NO_3_)_2_	11,350 ± 70	77 ± 1	6.11	DPt	<−8.750	>6.071	>14.821
PAN + Al(NO_3_)_3_	8160 ± 60	31 ± 2	1.53	DPt	−0.559	1.707	2.266
PAN	2530 ± 50	-	-	DGC	−1.554	1.628	3.182
PAN + LiNO_3_	7350 ± 60	8.8 ± 0.1	1.40	DGC	−1.544	1.644	3.188
PAN + Ca(NO_3_)_2_	21,620 ± 80	62 ± 1	2.09	DGC	−1.761	1.749	3.510
PAN + Mg(NO_3_)_2_	10,830 ± 70	77 ± 1	6.11	DGC	<−5.748	>6.255	>12.003
PAN + Al(NO_3_)_3_	8560 ± 60	31 ± 2	1.53	DGC	−0.251	1.648	1.899
PAN	-	-	-	SPE	−1.647	1.429	3.076
PAN + LiNO_3_	-	7.7 ± 0.5	1.20	SPE	−1.784	1.497	3.281
PAN + Ca(NO_3_)_2_	-	105 ± 1	4.60	SPE	−1.857	1.605	3.457
PAN + Mg(NO_3_)_2_	-	66 ± 1	4.5	SPE	<−2.038	>1.966	>4.004
PAN + Al(NO_3_)_3_	-	21.7 ± 0.4	0.97	SPE	−1.550	1.463	3.013

**Table 5 ijms-23-16040-t005:** Molal percentage influence of salt doping with maximum content studied of Li^+^, Ca^2+^, Mg^2+^ and Al^3+^ in the electrical conductivity (κ), density (ρ), viscosity (ƞ), refractive index (n_D_) and surface tension (σ) of EAN and PAN at 5, 25 and 95 °C.

PIL	Physical Property	Li^+^			Ca^2+^			Mg^2+^			Al^3+^		
		5 °C	25 °C	95 °C	5 °C	25 °C	95 °C	5 °C	25 °C	95 °C	5 °C	25 °C	95 °C
EAN	κ	−25.76	−21.15	−9.91	−60.82	−53.20	−34.60	−67.57	−58.98	−36.61	−86.05	−74.26	−42.95
EAN	ρ	2.55	2.52	2.57	-	-	-	6.48	6.41	6.34	7.55	7.44	7.39
EAN	ƞ	47.50	36.57	20.05	-	-	-	337.97	212.10	92.12	197.29	143.96	68.51
EAN	n_D_	-	0.21	-	-	-	-	-	0.42	-	-	0.62	-
EAN	σ	-	1.13	-	-	-	-	-	1.88	-	-	−4.02	-
PAN	κ	−27.76	−19.58	−10.62	-	−49.93 *	-	−21.04	−20.96	−16.37	−32.10	−27.19	−20.38
PAN	ρ	3.14	3.10	-	-	-	-	-	-	-	-	5.12	-
PAN	ƞ	45.62	34.68	-	-	-	-	-	-	-	-	120.66	-
PAN	n_D_	-	0.20	-	-	-	-	-	-	-	-	0.33	-
PAN	σ	-	0.46	-	-	-	-	-	-	-	-	−1.69	-

* In solid physical state.

**Table 6 ijms-23-16040-t006:** Materials employed, their supplier and their certified purity.

Material	Name	Supplier	Purity (%*w*/*w*)
EAN	Ethylammonium nitrate	IoLiTec (Heilbronn, Germany)	>97%
PAN	Propylammonium nitrate	IoLiTec (Heilbronn, Germany)	>97%
LiNO_3_	Lithium nitrate	Alfa Aesar (Ward Hill, Massachusetts, USA)	99.8%
Ca(NO_3_)_2_·4H_2_O	Calcium nitrate	PanReac AppliChem (Barcelona, Spain)	99.7%
Mg(NO_3_)_2_·6H_2_O	Magnesium nitrate	Fluka (Sigma-Aldrich) (St. Louise, Missouri, USA)	≥98.0%
Al(NO_3_)_3_·9H_2_O	Aluminum nitrate	Sigma-Aldrich (St. Louise, Missouri, USA)	99.7%
Fc	Ferrocene	Sigma-Aldrich (St. Louise, Missouri, USA)	99.9%
AgNO_3_	Silver nitrate	PanReac AppliChem (Barcelona, Spain)	99.8%

**Table 7 ijms-23-16040-t007:** Reduction formal potential (E’^0^) of the standard redox couple Fc-Fc^+^ and the applied effective area of the working electrode (A_WE_) in each IL and working electrode (WE).

IL	WE	E’^0^ (Fc-Fc^+^) (V)	A_WE_ (cm^2^)
EAN	SPE	0.075	0.419
EAN	DPt	−0.282	0.0432
EAN	DGC	−0.280	0.0347
PAN	SPE	0.038	0.233
PAN	DPt	−0.255	0.0351
PAN	DGC	−0.255	0.0301

## Data Availability

Data is contained within the article or Appendix A.

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
