# Peer review of "Influence of Metal Salts Addition on Physical and Electrochemical Properties of Ethyl and Propylammonium Nitrate"

_ijms, 2022, doi:10.3390/ijms232416040_

Round 1
Reviewer 1 Report
In current study, continues the characterization of the alkyl nitrate ILs by analysing the influence of inorganic nitrate salts addition on their physical properties which can simultaneously accomplish better knowledge on conductive materials properties.
In previous works, we presented the influence of small amounts of water on physical properties of the protic ionic liquids (ILs) ethylammonium nitrate (EAN) and propylammonium nitrate (PAN). – why is previous works stating important?
Novel results of testings have not been shown in abstract
Following questions have arisen:
1. The title should say something about novel result of the research and show the innovative result. This application is not definitely increasing the efficiency of the system alone. Limit the number of figures and tables, give only the most important one’s results. Three-line tables are preferred. Error bars are mandatory in figures.
2. „Capacity to combine an enormous number of cations and anions that makes them extraordinary“ The interpretation of literature and results needs clarifying with proper references. There could be enhanced the preciseness of used terms.
Various treatments to be solved for more economic ways using adsorbents or IL application, which could be shown: DOI: https://doi.org/10.3390/w13111522, https://doi.org/10.3390/w13141969, https://doi.org/10.3390/w14020242, https://doi.org/10.3390/w13152136 , https://doi.org/10.3390/w13081095
3.The choice of EAN and PAN chosen concentrations needs clarifying. Are there other similar ones potentially be used that can work better? How did literature stated? What were the measures in J, viscosity (ƞ), electrical conductivity (κ), density (ρ), surface tension (σ) and refractive index 82 (nD) with water concentration in other references?
Hypothesis and aims are missing from current MS, need to add these. Novelty aspects as well. Introduction section cannot state Your own results already as this could be included in results part.
5. In my opinion, main point of this study was somehow missed providing extensive discussion on too many details on bacterial data of characteristics, but not about main aims pointed.
6. Language and structuring of the work should be substantially revised.
7. There should be some specific conclusions and the main point of the work better introduced.
Without concrete values and more sophisticated statistical techniques the text of the abstract and other sections remains vague. Word order of the sentences needs revision.
Abbreviations in the manuscript body, at the first occurrence, should be in abbreviated form plus full definition; then they should be given only in abbreviated forms throughout the manuscript.
Results
8.Some general sentences need to be adjusted in exact numbers and all limits.
The novelty of experimental set-up and operational strategy should be presented in a more clear way. Comparisons with other studies performed should be included.
9.What was exact significance level, p values?
Please provide the standard deviations for the operational parameters of the system presented in this chapter.
- Make sure same statements hold which are in the same order as in abstract and results. Results should be shown in numerical manner in conclusion and abstract
References
The references style should be made consistent (i.e. Journals names consistently with capital letters, remove commas before the authors names before last author), use automatic citation program.
Author Response
Please, see the attachment.

Reviewer 2 Report
The paper titled “Influence of metal salts addition on properties of ethyl and propylammonium nitrate” reports how the presence of monovalent (Li+), divalent (Ca2+ and Mg2+) and trivalent (Al3+) metal cations influences the five physical properties (viscosity, electrical conductivity, density, surface tension and refractive index). The paper is perfectly written from scientific point of view. Therefore, with a pleasure I recommend it for publication. Some technical corrections required to make the paper more favorable for the readers are listed below.
1. Figures 1-7: The color-defining legend should be used instead of the description in the captions.
2. Line 163: “2.2. Electrochemical of the PILs samples” -?-> “2.2. Electrochemical properties of the PILs samples”
3. Line 214: “changes respect the temperature gradient” -?-> “changes with respect to the temperature gradient”
4. Line 220, 249, 476: “in the physical properties” -> “on the physical properties”
5. Line 386: “LINO3” -> “LiNO3”
6. Line 468: “To sum up, a physical and electrochemical characterization was made” -> “To sum up, physical and electrochemical characterizations were made”
7. Line 489: “the effect of metal cations in surface tension” -> “the effect of metal cations on surface tension”
8. Data Availability Statement is inappropriate.
Author Response
Please, see the attachment.
